# Evolution of Multicellular Complexity in The Dictyostelid Social Amoebas

**DOI:** 10.3390/genes12040487

**Published:** 2021-03-27

**Authors:** Koryu Kin, Pauline Schaap

**Affiliations:** 1School of Life Sciences, University of Dundee, Dundee DD1 5EH, UK; koryu.kin@upf.edu; 2Institut de Biologia Evolutiva (CSIC-Universitat Pompeu Fabra), Passeig Marítim de la Barceloneta 37–49, 08003 Barcelona, Spain

**Keywords:** evolution of multicellularity, amoebozoa, dictyostelia, cAMP signalling, encystation, cell type evolution

## Abstract

Multicellularity evolved repeatedly in the history of life, but how it unfolded varies greatly between different lineages. Dictyostelid social amoebas offer a good system to study the evolution of multicellular complexity, with a well-resolved phylogeny and molecular genetic tools being available. We compare the life cycles of the Dictyostelids with closely related amoebozoans to show that complex life cycles were already present in the unicellular common ancestor of Dictyostelids. We propose frost resistance as an early driver of multicellular evolution in Dictyostelids and show that the cell signalling pathways for differentiating spore and stalk cells evolved from that for encystation. The stalk cell differentiation program was further modified, possibly through gene duplication, to evolve a new cell type, cup cells, in Group 4 Dictyostelids. Studies in various multicellular organisms, including Dictyostelids, volvocine algae, and metazoans, suggest as a common principle in the evolution of multicellular complexity that unicellular regulatory programs for adapting to environmental change serve as “proto-cell types” for subsequent evolution of multicellular organisms. Later, new cell types could further evolve by duplicating and diversifying the “proto-cell type” gene regulatory networks.

## 1. Introduction

Multicellularity emerged in an early stage of the history of life. The oldest fossil record retaining the trace of putative multicellular organisms goes back to 2.1 billion years ago [1,2]. Since then, multicellularity evolved many times independently in all domains of life [3,4,5,6,7] and can even be evolved in the laboratory within a short period [8,9,10]. Although the evolution of multicellularity itself is recurrent, the way in which multicellular life evolved varied greatly between different lineages. Some lineages have adhered to a simple form of multicellularity without much cell differentiation, while others gradually evolved to divide the labours between cells in the multicellular body [11]. Some have abandoned multicellularity altogether [12], while others have evolved to become gigantic multicellular organisms composed of more than 10^13^ cells, which are are conscious of their own existence in the universe.

In discussing the evolution of multicellularity, we regard “multicellular complexity” as a concept in continuum. We are aware that many workers treat “simple multicellularity” and “complex multicellularity” in a dichotomous way, as if all multicellular organisms clearly fell into either category. They restrict the use of the term “complex multicellularity” to certain groups with obligate multicellularity, where multiple cell types are organised in a three-dimensional pattern (primarily plants, fungi, and animals) [5,13], but we would like to take a different approach, as we believe the dichotomy does not capture the reality of nature. Even in the multicellular lineages which are often considered “simple,” a clear progression in morphological and behavioural complexity is often observed (e.g., in the volvocine algal lineage [14,15] or in the Dictyostelid social amoebas, as described below). Moreover, even within “complex” multicellular groups, organisms differ greatly in complexity [16,17,18]. It has been argued that the concept of “individuality,” a hallmark of complex multicellular organisms, cannot be captured in a dichotomous manner, either [15,19].

The presence of independently evolved lineages of multicellular organisms with different levels of complexity provides a unique opportunity to tease apart chance and necessity in the evolution of multicellular complexity. We have a number of “parallel worlds” or natural evolutionary experiments starting from different initial conditions. In principle, genomic, sociobiological, and ecological factors could all influence the subsequent evolution of multicellular complexity. For instance, in some lineages, certain genomic features, such as large intron size, might have contributed to the increase in multicellular complexity [13]. Sociobiological factors, such as the conflicts between genetically nonhomogeneous cells constituting the body, could have imposed a constraint for the evolution of complexity in aggregative multicellular organisms [20] (but see also [21,22] for the diversity of aggregative multicellular organisms). Additionally, environmental factors, such as atmospheric oxygen concentration [23,24] and global glaciation [25,26], as well as ecological factors, such as the presence of predators/preys [10,27], may all have played important roles in the evolution of multicellularity.

Despite huge differences in the ways different groups of multicellular organisms evolved, we believe that some common patterns can be recognised in the evolution of multicellular complexity. In this review, we will explore the evolution of multicellular complexity in the Dictyostelid social amoebas with the aim of presenting some commonly observed patterns in multicellular evolution with specific examples. We will present evidence that major Dictyostelid cell types, spore and stalk cells, evolved through the modification of the regulatory programs for encystation, a unicellular survival strategy. We will also discuss our ongoing effort to understand the evolution of novel cell types in the Dictyostelids. We will conclude by pointing out growing evidence in various multicellular lineages that ancestral unicellular phenotypes, which are temporarily activated through environmental stimuli, serve as “proto-cell types” from which specialised cell types in multicellular organisms evolved. This could initially occur by bringing the expression of specialised functions from temporal into spatial control [28,29] and by refining and enhancing the specialised roles. Subsequently, multicellular complexity could increase through the evolution of a novel cell type by duplication of a gene regulatory network, followed by divergence.

## 2. A Parallel World of Multicellular Complexity: The Evolution of Social Amoebas

### 2.1. Social, Sexual, and Solitary Life Cycles of the Social Amoebas

Dictyostelid social amoebas consist of about 150 known species. Their phylogenetic relationships were first reconstructed with molecular data based on SSU rDNA, which established four major groups of Dictyostelids, simply named Groups 1, 2, 3, and 4 [30]. Since then, the phylogeny has been revised through phylogenomic studies [31,32,33], but the major groupings stayed the same. Groups 1, 2, 3, and 4 were recently reclassified as Cavenderiaceae, Acytosteliaceae, Raperosteliaceae, and Dictyosteliaceae [34], but there are still disagreements with respect to the relationships between a few groups [32], which may lead to further renaming in the near future. In this article, we adhere to the genus names which have been traditionally used, as well as “Group 1–4” for referring to major groups.

All Dictyostelids share some common features in their life cycle (Figure 1A). In the vegetative cycle, they proliferate as unicellular amoebas for as long as enough of their bacterial food is available. Upon starvation, they start to secrete a chemoattractant and form a multicellular aggregate of up to 100,000 cells in *Dictyostelium discoideum*. This aggregate then assumes a sausage shape, called the sorogen. In some species, sorogens migrate as slugs to find a spot favourable for fruiting body formation. The sorogens then undergo culmination into fruiting bodies (sorocarps), which in most species consist of two cell types, spores and stalk cells. Stalk cells are vacuolated cells with a sturdy cell wall, which die after being encased into a cellulose stalk tube. Spore cells are carried aloft and also become encased in a wall. They enter dormancy until the environmental conditions become favourable for survival as amoeba. Once the conditions improve, which may happen when spores are being carried to a new environment by rain or passing insects, spores germinate and restart a new life cycle.

In addition to the vegetative and multicellular cycles described above, the sexual cycle and the encystation cycle have also been observed in many species. Like the asexual multicellular cycle, the sexual cycle commences upon nutrient depletion but is favoured over multicellular development in dark and humid or submerged conditions [37]. At the beginning of the sexual cycle, mating of haploid cells occurs. There are three mating types known in *D. discoideum*, where each of the three mating types can mate with each of the other two but not with itself [38]. Since there is no mechanism to prevent fusion between sexually compatible cells after the initial fertilisation event, multiple haploid cells can fuse to create multinucleate syncytia. Syncytia undergo sequential cytokinesis until only two nuclei remain, at which point two nuclei fuse to form a diploid nucleus. Diploid cells, called “giant cells,” attract surrounding haploid cells by secreting cAMP. Giant cells then phagocytose the aggregating cells clinging to them, while the aggregating cells located outside produce an outer wall, encasing them. Once giant cells eat up all the cells within the wall, they enter into a period of dormancy, known as “macrocysts.” Diploid nuclei undergo meiosis, followed by multiple mitoses, to produce multiple haploid nuclei. Finally, when conditions still unknown are met, macrocysts germinate, and haploid amoebas crawl out of them. The presence of the sexual cycle has been confirmed in Dictyostelid species across all major groups, and it is thus considered to be an ancestral trait for this group.

Unlike the multicellular cycle and the asexual cycle, which involves aggregation of multiple cells, the encystation cycle in the Dictyostelids occurs unicellularly. Encystation is a unicellular stress response observed throughout eukaryotes to produce cysts, a resistant cell type encapsulated in a cellulose- or chitin-rich wall [39]. In many Dictyostelid species (but not in Group 4 species, including *D. discoideum*), amoebas differentiate into cysts (sometimes also called “microcysts” in order to contrast it with “macrocysts” produced in the sexual cycle) when starved in darkness or under high osmolarity or submerged conditions. Once the environmental conditions improve, the cysts germinate to release amoebas. The presence of the encystation cycle in non-Group 4 species, as well as in other Amoebozoan species, suggests that it is an ancestral trait for Dictyostelia but was lost in the Group 4 Dictyostelids.

### 2.2. “Social amoebas” in Amoebozoa—The Phylogenetic Context

The Dictyostelids belong to the phylum Amoebozoa, which consists of eukaryotes with amoeboid or amoeboflagellate cellular morphology [40,41]. Amoebozoa is in turn a member of the phylogenetic supergroup Amorphea, together with Holozoa (Metazoans + unicellular relatives) and Fungi [42] (Figure 2). To retrace the emergence of Dictyostelid multicellularity, it is essential to know the characteristics of the Amoebozoan Unicellular Common Ancestor of Dictyostelids (AUCAD).

The extant amoebozoan group closest to the Dictyostelids is the myxomycetes, which consists of three groups: Ceratiomyxa, Protosporangiid, and Myxogastria. Myxogastrids are the most species-rich group in the myxomycetes and are also known as plasmodial slime moulds. They have complex life cycles, which consist of unicellular and plasmodial phases [35,43] (Figure 1B). In the unicellular phase, they feed on bacteria, while either in amoeboid or in flagellate form (collectively called “amoeboflagellate”), depending on the environmental conditions. Amoebas (walkers) are favoured in dry environments, while flagellates (swimmers) are favoured in moist or wet environments. When either form is under adverse environmental conditions, it differentiates into a resistant cyst and remains dormant until conditions improve. Once amoeboflagellate cells of compatible mating types meet, cell fusion occurs, and the plasmodial cycle starts [44]. Karyogamy of two nuclei creates diploid zygotes, and nuclear divisions without cytokinesis create a multinucleate plasmodium, which moves around and grows by feeding on other cells in the environment. Under suitable conditions, sporulation occurs to produce haploid spore cells by meiosis and cytokinesis. The spores are located on top of stalks, which are constructed from the cytoplasmic cell mass. In favourable conditions, spores germinate to restart the cycles.

When the life cycles of Dictyostelids and myxogastrids are compared, the presence of diploid zygotes makes it reasonable to infer that the plasmodial cycle of myxogastrids is comparable to the sexual cycle of Dictyostelids, rather than to the multicellular cycle (see [45,46] for more discussions on this point).

Going further back in the tree of life, the closest relatives of Dictyostelids and myxomycetes are Archamoeboae. They include a variety of anaerobic protists with both free-living and endobiont members. Although their exact life cycles are largely uncharacterised, their cellular phenotypes are polymorphic [47]. Many members of Archamoebae studied to date, e.g., *Mastigamoeba*, can produce some or all of the following cell types: flagellates, aflagellate amoebas, large multinucleate amoebas, and cysts [36,47,48]. Parasitic members, e.g., *Entamoeba*, have lost the flagellar apparatus but can still undergo encystation, which is an essential stage of their life cycle. Large multinucleate amoebas seen in this group are reminiscent of the syncytial plasmodia formed in myxogastrids or of the multinucleate cells, which often occur at the beginning of the Dictyostelid sexual cycle, but there is no evidence that multinucleate amoebas are related to sexual processes. In fact, the sexual cycle of Archamoebae is virtually unknown, although the presence of many genes related to sex in their genomes suggests that it is present [49,50].

Comparing the life cycles of these related groups with that of Dictyostelids, it becomes clear that the AUCAD already had rather complex life cycles. The AUCAD most likely had at least three cellular forms: amoebas, flagellates, and cysts. In addition, it almost certainly had a sexual cycle that involved multinucleate amoebas. The flagellate form was lost during the evolution of Dictyostelids, but other forms persisted with modification.

### 2.3. Ecological Advantages of Spores over Cysts: Increased Dispersal or Frost Resistance?

If the AUCAD already had cysts, an environmentally resistant cell type that is produced unicellularly, what could be an evolutionary advantage of making another resistant cell type, spores, through multicellular development? In order to answer this question, we need to understand the environmental and ecological settings where social amoebas evolved. Knowledge of the divergence time of Dictyostelia is extremely important in this context.

Previous molecular clock estimations of the divergence time of the two major branches of Dictyostelia varied widely, ranging from 330 million years ago (mya) to 650 mya [51,52,53], reflecting the difference in the datasets and the methods used. The most recent estimate, which is based on relaxed molecular clock dating, using an alignment of 240 proteins with new fossil calibration points of testate amoebas, lies shortly after the start of the Cambrian period at 519 ± 140 mya (Figure 2) [26]. Albeit with a large margin of error, the estimate suggests that the divergence of Dictyostelids coincides with the famous Cambrian radiation of metazoans (although it is thought that most metazoan phyla already existed well before Cambrian).

A major ecological adaptation of fruiting body formation, which was put forward previously, is its advantage for dispersing spores through contact with motile carriers, especially small invertebrates in the soil [54]. Indeed, laboratory experiments showed that fruit flies exposed to intact fruiting bodies of *D. discoideum* carried significantly more spores than those exposed to disrupted ones [55]. However, if social amoebas had already evolved in the Cambrian period, where there were no invertebrates on land to disperse them, increased dispersal cannot be the initial adaptation that triggered the evolution of multicellularity.

Recently, another potential ecological advantage of spores over cysts has been suggested: frost resistance. By examining the survival curves of cysts and spores from 29 Dictyostelids from all major groups, Lawal and co-authors showed that spores withstand frost better than cysts, with Group 4 spores being the most frost resistant. In nonfreezing temperatures, survival rates between spores and cysts were not significantly different [26]. Comparative ultrastructural studies of spores and cysts revealed that spores are more compacted than cysts and have well-defined three-layered walls, while cyst walls consist of only one or two layers. In addition, Group 4 species combined thick walls with the highest level of compaction. Strikingly, Group 4 species are often found in arctic and alpine regions, which is not the case for species in Groups 1–3, indicating that these adaptations to their spores allowed Group 4 to colonise colder habitats. As mentioned earlier, the fossil-calibrated phylogeny of Amoebozoa sets the split between the two major branches of Dictyostelia at 0.52 billion years ago (Figure 2), following the global glaciations known as “snowball earth” [56,57]. Combined, these observations suggest that an adaptation to cold climate has been a driving force for the evolution of sporulation in Dictyostelia. We can recognise two major steps here: the first step was during the initial phase of Dictyostelid evolution, which triggered the evolution of multicellular spore formation, and the second step was during the dispersal of Dictyostelids to colder habitats [26].

While this explains the evolution of spores, it does not explain the differentiation of stalk cells. The purpose of the stalk is thought to aid spore dispersal by lifting the spores, but this is unlikely to be the whole story. In *D. discoideum*, the prespore cells secrete DIF-1 [58], which induces other cells to die by extreme autophagy, while differentiating into stalk cells and passing through the prespore cell mass. It is well possible that the metabolites produced by this sacrifice are taken up by the prespore cells and used to synthesise their thick spore walls. As further discussed in the next paragraph, unicellular encystation solely requires an increase of intracellular cAMP and activation of cAMP-dependent protein kinase (PKA) [59], while spore differentiation additionally requires an increase in extracellular cAMP and activation of cell surface cAMP receptors (cARs) [60]. Because *D. discoideum* amoebas secrete most of the cAMP that they produce [61], extracellular cAMP naturally increases once cells are close together in aggregates. High extracellular cAMP can therefore be considered as a signal for the aggregated state. When cAR genes are deleted in *Polysphondylium pallidum*, it will form cysts in its fruiting bodies instead of spores [60]. cAR-mediated induction of prespore gene expression is specifically inhibited in *D. discoideum* mutants with defective autophagy [62]. In contrast to the stalk cells, the prespore cells do not actually show much autophagy [63]. It therefore appears that Dictyostelid amoebas will form spores only if the stalk population undergoes autophagy. In this scenario, the stalk cells not only function to lift the spores but also to feed them, typically a major role for somatic cells in multicellular organisms.

### 2.4. Evolution of Spore and Stalk Differentiation Pathways from an Ancestral Encystation Pathway

Even though the AUCAD already had complex life cycles, the multicellular cycle does appear to be an evolutionary innovation in Dictyostelia. The question is whether the genetic program for multicellular development evolved more or less “from scratch” through a large-scale innovation in their ancestral genome or if we can trace some of the components back in evolution. Now, we will briefly look at the molecular mechanisms that regulate multicellular development in the model species *D. discoideum.* For comprehensive reviews of these mechanisms, see [64,65]. Here, we will focus on only those aspects that are pertinent to our current discussion about the evolution of multicellularity.

The most prominent feature of *D. discoideum* development is its heavy reliance on cAMP signalling. The first identified role for cAMP was its function as chemoattractant for aggregation, which later was found to be an evolutionary innovation specific to the Group 4 [66]. As briefly discussed in the previous section, cAMP has evolutionarily more ancient roles in spore and stalk cell differentiation, which are derived from an ancestral role of cAMP as signal for stress-induced encystation (Figure 3A). In this role, as established in the encysting Dictyostelid *P. pallidum,* drought and starvation stress activate the adenylate cyclases AcrA and AcgA to synthesise cAMP, which, by activating PKA, triggers encystation [59,67]. The cAMP phosphodiesterase RegA, itself activated by sensor histidine kinases (SHKs) and inactivated by sensor histidine phosphatases (SHPs), inhibits encystation and promotes excystation by hydrolysing cAMP [68]. PKA, AcrA, RegA, and large numbers of SHK/Ps are conserved throughout Amoebozoa [51,69,70] and, except for AcrA, also in Excavates [71]. RegA also inhibits encystation of the unicellular Amoebozoan *Acanthamoeba* [68], making it likely that this cAMP-mediated stress response is conserved throughout at least Amoebozoa.

This linear stress response bifurcated to control both the differentiation of spore and stalk cells during multicellular development (Figure 3B,C). Here, the SHK/Ps came to play crucial roles in intercellular communication. Like cysts, the spores and stalk cells are encapsulated in a rigid cell wall. Since the ability of Dictyostelids to aggregate and form fruiting bodies depends on the motility of its amoebas, it is essential that spores and stalk cells mature at the correct time and place. The spore walls are, therefore, prefabricated in Golgi-derived prespore vesicles just after aggregation, ready for rapid assembly after almost instantaneous exocytosis in late fruiting body formation. Spore coat gene expression is triggered by an AcgA-induced increase in both intra- and extracellular cAMP levels in the posterior region of the slug, followed by activation of both cARs and PKA [72,73,74]. The prespore cells, in turn, release DIF-1 and possibly other factors that antagonise prespore differentiation [58]. This sets up a proportion of non-prespore anterior-like cells (ALCs) that move towards the front and rear of the slug to later form the stalk, basal disc and other support structures, known as the upper and lower cup (Figure 3B).

The transition from slug migration to fruiting body formation is inhibited by ammonia, which is produced by autophagy in the starving cells [75]. Ammonia activates the SHK DhkC to phosphorylate and, thereby, activate RegA to hydrolyse intracellular cAMP and, thereby, block further PKA activation [76]. Aerial projection of the slug tip in response to incident light allows dissipation of ammonia away from the tip, thus relieving the block. Cross talk between morphogenetic control and cell differentiation on one hand and between maturing stalk and spore cells on the other further regulate fruiting body formation (Figure 3C). Both aggregation and slug and fruiting body morphogenesis are coordinated by cAMP waves that are produced by the adenylate cyclase AcaA [77,78]. In slugs, AcaA is predominantly expressed at the tip [79], the organiser of morphogenesis, and the site where stalk formation initiates. The anterior prestalk cells secrete c-di-GMP, the inducer of stalk maturation [80], which hyperactivates AcaA at the tip to produce cAMP, activate PKA, and trigger stalk gene expression [81]. Meanwhile, the prespore cells secrete a protein AcbA, which is cleaved by a protease, TagC, that is expressed on the surface of prestalk cells to produce the peptide SDF-2 [82]. SDF-2 next activates the SHP DhkA on prespore cells, which dephosphorylates and, thereby, inhibits RegA, allowing PKA to be activated [83]. Active PKA then triggers spore maturation by exocytosis of the prespore vesicles and expression of some spore-specific genes.

In addition to the cAMP pathway, there are many other genes involved in Dictyostelid development and cell differentiation. Based on published experiments, a list was compiled of 385 genes, which were reported to be essential for normal developmental progression in *D. discoideum.* The presence of these 385 genes was studied in Dictyostelid genomes as well as in the genomes of unicellular Amoebozoa [70,84]. Nearly 80% of these developmentally essential genes are present in non-Dictyostelid amoebozoan species. The remaining 20% of genes, specific to Dictyostelids were enriched in transmembrane proteins or proteins secreted extracellularly. This implies that most of the intracellular machinery for cellular differentiation was already present in the AUCAD. What emerged during the evolution of multicellularity were the modules for cell–cell communication, such as secreted signals and their receptors and cell adhesion proteins. Another interesting point was that a few genes for synthesising important nonpeptide signalling molecules in *D. discoideum*, such as discadenine, c-di-GMP, and DIF-1, were found to be the products of lateral gene transfer (LGT) from bacterial species. DIF-1 is a chlorinated alkyl phenone made from a 12-carbon polyketide [85], and it is required to induce another cell type, the basal disc cells that form a support structure for the main stalk [86,87,88]. It is worth noting that rampant LGT was also found in unicellular (or facultatively multicellular) relatives of metazoans [89,90], which suggests that LGT could play an important role in providing cells in early multicellular organisms with new means to communicate with each other.

### 2.5. Major Developmental Innovations and Cell Type Evolution in Group 4 Dictyostelids

Once the basic program for multicellularity was established in Dictyostelia, it further evolved and diversified in different lineages. Mapping of 24 morphological and developmental traits onto a molecular phylogeny of 99 Dictyostelid species, followed by the reconstruction of ancestral states, revealed that the last common ancestor (LCA) to Dictyostelia formed relatively small and clustered fruiting bodies, consisting of a stalk and apical spore mass [91]. This organism likely used the dipeptide glorin, and not cAMP, as the attractant for aggregation and could still encyst. Its slugs showed little or no migration and no pattern of prespore and prestalk cells. Instead, all cells initially differentiated into prespore cells, with a fraction re-differentiating into stalk cells after having reached the tip. This phenotype was mostly retained throughout Groups 1 and 3. In Group 2, one clade lost the cellular stalk and only produced a cellulose tube to carry the spores, while, in the other clade, fruiting structures developed regular whorls of side branches. However, this feature was not unique to Group 2 and also evolved in a small sister clade to Group 4, while other branching patterns also evolved multiple times independently across Groups 1, 2 and 3 [91,92].

Major innovations occurred in the LCA to Group 4. It started to use cAMP as attractant and formed larger, solitary, and unbranched fruiting bodies. It also showed extensive slug migration, set aside a population of non-prespore cells for later differentiation into supporting cells, and evolved new cell types, called cup cells and basal disc cells. As highlighted before, its spores became more frost resistant [26]. It is not yet clear whether some of these changes are correlated and have a common genetic basis. However, we are beginning to uncover the molecular basis of cell type evolution, especially of the evolution of cup cells, in Dictyostelids. We will briefly describe their basic characteristics, as well as some of the recent discoveries.

Basal disc cells are similar to stalk cells in being heavily vacuolated and surrounded by cellulose wall. They form a disc-shaped structure at the base of the stalk but are, unlike the stalk cells, not encased in a cellulose stalk tube. Cup cells are amoeboid cells that are located at the top and bottom part of the ascending spore mass during culmination (see Figure 1A). Developmentally, cup and basal disc cells are both derived from cell populations called anterior-like cells (ALCs). ALCs were initially identified by their propensity to be stained with neutral red, like prestalk cells, due to a preponderance of acidic vesicles in the cells [93,94,95]. However, unlike prestalk cells, which are located in the anterior part of the migrating slugs, ALCs are scattered in the posterior region where prespore cells form the majority. They later move both upwards and downwards out of the prespore region. The upward moving cells end up at the prespore/prestalk boundary, where they can either become incorporated in the prestalk region and stalk, as the prestalk cells become depleted during stalk formation, or they can form an upper cup to the spore mass. The downward moving cells form either a basal disc to support the stalk or a lower cup at the base of the sorus [93,94,95]. The function of the cup cells is not completely understood, but they may serve to elevate the spore head, as the ablation of cup cells, especially upper cup cells, causes the spore head to get stuck in the middle of the stalk without moving up [96]. Many prestalk cell markers, such as *ecmA* and *ecmB*, are not only expressed in prestalk cells but also in ALCs, often from distinct promoter elements [97,98,99]. Basal disc cells are structurally similar to stalk cells, but cup cells remain amoeboid throughout development. The amoeboid phenotype of cup cells represents a true evolutionary novelty in Dictyostelid multicellularity, in the sense that it is a clear departure from the cyst-like, walled phenotypes of the spore, stalk, and basal disc cells.

Recently, several genes upregulated by higher c-di-GMP concentrations than stalk genes were found to be specifically expressed in cup cells [81]. Unlike *ecmA* and *ecmB*, these genes are typically expressed at the final stage of fruiting body formation, after the (pre)spore mass has already risen to the top of the stalk. This suggests that cup cells may have other functions besides lifting the spore head. Utilising a late cup cell marker gene, *beiB*, cup cells were purified, and RNA-Seq data were compared with other terminally differentiated cells [100]. It appeared that the transcriptome of cup cells is most closely related to that of stalk cells, which is consistent with the developmental origin of cup cells and prestalk cells from ALCs. However, 842 genes specific to cup cells were also identified. This set was enriched in small GTPases involved in cell adhesion, cell movement, or taxis. In addition, Group 4-specific small peptides in the hssA/7E/2C gene family [101,102] were highly expressed in cup cells as well as some transcription factors. One of them, a transcription factor in the cud-like gene family called *cdl1a*, proved to be essential for cup differentiation [103]. Interestingly, this transcription factor evolved through a gene duplication event specific to Group 4 Dictyostelids [103,104], while the proto-orthologue of *cdl1a* ancestrally functioned in the regulation of stalk formation. These findings suggest the importance of gene duplication for the evolution of a novel cell type in the Dictyostelids.

## 3. Conclusions

Traditionally, the evolution of multicellularity has been often thought to follow a linear progression from undifferentiated clusters of cells to morphologically complex forms with differentiated cells. We can perhaps call this a Haeckelian view, because it is highly influenced by Haeckel’s idea of recapitulation [105]. We consider this view as too simplistic, if not entirely wrong, because the unicellular ancestors of multicellular organisms were unlikely to be as simple and blank as the Haeckelian view assumes.

Instead, we argue that unicellular ancestors often already had complex life cycles with discrete adaptive phenotypes. These discrete phenotypes, typically regulated by environmental stimuli, serve as “proto-cell types” from which multicellular developmental programs crystalise. As discussed above, in the evolution of Dictyostelia, encystation, an ancestral unicellular survival mechanism, provided the core molecular mechanism from which the multicellular cell differentiation program evolved. Similar cases can be found in other multicellular lineages, as well. For instance, in the volvocine algae, it was found that *regA*, a gene-controlling somatic cell differentiation in *Volvox carteri*, evolved from *rls1*, a gene that controls stress response in *Chlamydomonas reinhardtii*, a unicellular relative of *Volvox* [29,106,107]. In metazoans, genome analysis of many unicellular relatives of animals, such as choanoflagellates, filastereans, and icthyosporeans, revealed that many of the genes essential for metazoan development were already present in their unicellular relatives [90,108,109,110]. These “developmental” genes were possibly used for temporal regulation of phenotypic change in the course of their life cycles [6,111].

The next step in multicellular evolution could be spatial reorganisation of the “proto-cell types,” through the actions of cell–cell signalling [6]. The proto-cell types might have been refined to become mature cell types over time, with the suppression of unnecessary genes and the evolution of proteins for more specific functions. In the Dictyostelids, evolutionary transitions from encysting “proto-cell type” to fully fledged spore and stalk cells required the acquisition of signalling molecules and some receptors through LGT [84] and the adaptation of other receptors, especially sensor histidine kinases, to novel ligands. By evolving the new cell–cell signalling modules, the Dictyostelid common ancestor could assimilate the environmental signals into the genome: i.e., the cell differentiation programs which were previously triggered only by environmental stimuli can now be regulated by signals that are exchanged between the cells, for which the biosynthetic machinery is now genetically encoded. Evolution of new cell types by assimilating plastic responses to environmental stimuli has also been suggested in other contexts [112,113,114,115] and might be a common mechanism for the evolution of complexity.

In Group 4 Dictyostelids, new cell types evolved, along with other developmental innovations. Transcriptomics revealed that a novel cell type, the cup cells, evolved by modifying the stalk cell gene regulatory network. In addition, the available data suggest that duplication and divergence of a transcription factor, which ancestrally functioned in the stalk gene regulatory network, played a crucial role in the evolution of cup cells. The duplication and divergence of gene regulatory networks has also been suggested as a major mechanism for the evolution of new cell types in metazoans [116,117], and it may well be another common mechanism through which multicellular complexity evolves.

An issue that we did not discuss is the sociobiological aspect of the evolution of multicellularity. Dictyostelia become multicellular by aggregation, and the cells contributing to the fruiting body are not necessarily genetically identical, which may give rise to genetic conflict [20]. However, in spite of such conflict, there has been a clear increase in morphological complexity in the Group 4 Dictyostelids, as we have seen, although over their 500 million years of evolution, progress towards complexity has been relatively modest. The issue of genetic conflict has been studied most extensively in *D. discoideum* [118,119]. A high level of relatedness appears to have been maintained in this species [120], and a mechanism for kin discrimination within the aggregate, which involves polymorphic *Tgr* genes, has been discovered [121,122]. In addition to molecular mechanisms to maintain high relatedness, another factor that may limit the impact of genetic conflict is the fact that cells spend only one generation as multicellular spores but can go through hundreds of generations as feeding amoebas before aggregation. Competition for food or light, a major driving force for metazoan and plant evolution, will, therefore, in Dictyostelia, not act on the multicellular state, perhaps simultaneously explaining its slow evolution in terms of morphological complexity and the limited challenge posed by genetic conflict.

## Figures and Tables

**Figure 1 genes-12-00487-f001:**
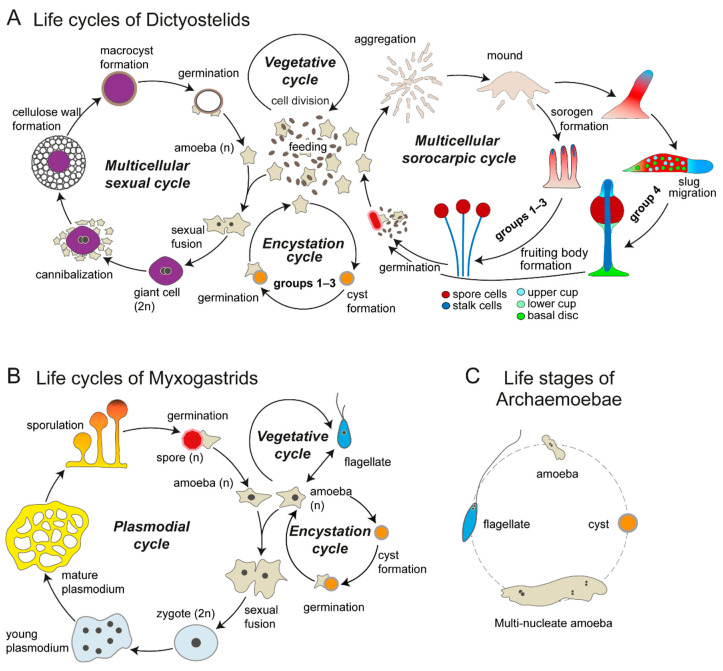
Life cycles of Amoebozoan groups related to the Dictyostelids. (**A**). The Dictyostelids show four different cycles, the unicellular vegetative and encystation cycles and the multicellular sexual and sorocarpic cycles. The depiction of the sorocarpic multicellular cycle shows generalised differences between Dictyostelids in Groups 1–3 and those in Group 4. Different cell types in the fruiting body are shown in different colours. Cup cells and basal disc cells are found only in Group 4, while encystation is lost in this group. (**B**). The Myxogastrids show three major life cycles, vegetative, plasmodial (or sexual), and encystation, as depicted here. In addition, not shown here, the plasmodium can turn into a dormant dehydrated structure, called sclerotium, in harsh environmental conditions, and can also be produced with haploid cells (apogametic cycle). Redrawn after Figure 5 in [35]. (**C**). In many species of Archaemoeba, multiple cellular phenotypes, such as amoeba, multinucleate amoeba, cyst, and flagellate are known to exist, but there is not much information about how they transition from one to another. Redrawn after Figure 1 in [36].

**Figure 2 genes-12-00487-f002:**
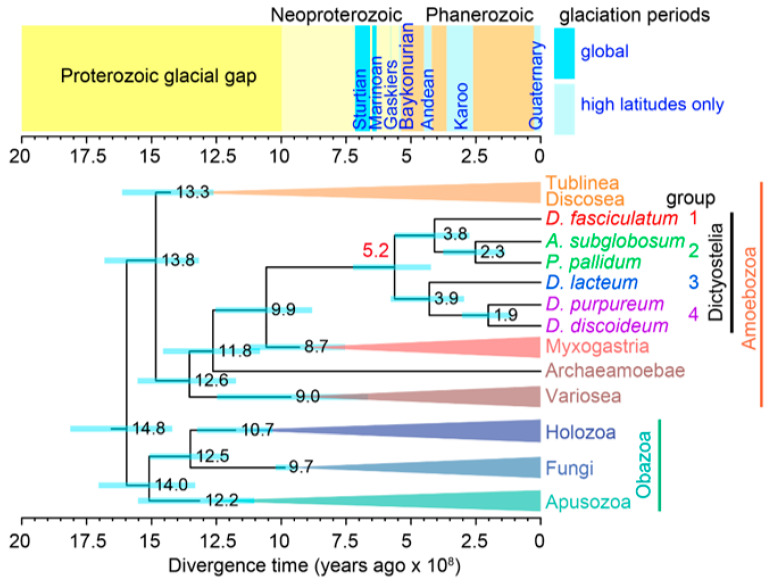
**Time tree of Amoebozoa.** A. Simplified tree of Amoebozoa with a few related groups of eukaryotes, representing the relationships of major groups. The positions of the 4 major groups of Dictyostelia are indicated. Divergence time is shown in 100 million years. Blue bars on the nodes represent Bayesian 95% Highest Posterior Density (HPD) intervals. Major global glaciation periods are shown above the tree. Redrawn from Figure 5 in [26].

**Figure 3 genes-12-00487-f003:**
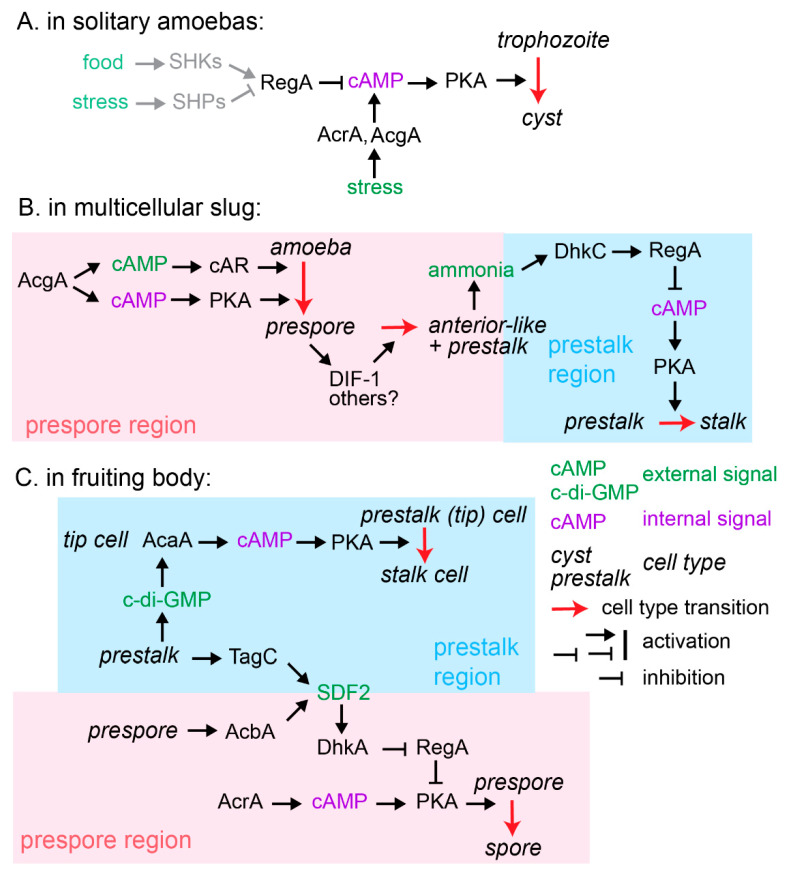
Dictyostelid signalling pathways evolved from an amoebozoan stress response. (**A**). In solitary amoebas, starvation or drought stress increases intracellular cAMP levels to activate (PKA), which induces the transition from growing trophozoite into walled dormant cyst. The cAMP phosphodiesterase, RegA, inhibits this process by hydrolysing cAMP; it bears a conserved response regulator domain that is the target for phosphorylation/dephosphorylation by sensor histidine kinases/phosphatase (SHK/Ps) that, respectively, activate/inactivate the hydrolytic activity. The signal module of SHK/Ps, RegA, PKA, and AcrA is conserved throughout Amoebozoa. (**B**,**C**) In the multicellular Dictyostelids, this stress pathway came under control of signals exchanged between the cells that regulate the differentiation of walled spores and stalk cells. See main text for details. Other abbreviations: AcrA, AcgA, AcaA: adenylate cyclases R, G, and A; PKA: cAMP-dependent protein kinase; cAR: cell surface cAMP receptor; DhkC: sensor histidine kinase C; DhkA: sensor histidine phosphatase A; TagC: tight aggregate C—a protease; AcbA: acetyl-coA-binding protein A, the precursor of SDF2: spore differentiation factor 2—a peptide; DIF-1: differentiation factor 1—a chlorinated polyketide; c-di-GMP: 3’,5’-cyclic diguanylic acid.

## Data Availability

Not applicable.

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
