# Peer review of "Evolution of Multicellular Complexity in The Dictyostelid Social Amoebas"

_genes, 2021, doi:10.3390/genes12040487_

Round 1

Reviewer 1 Report

In the Genes review manuscript entitled "Evolution of Multicellular Complexity in the Dictyostelid Social Amoebas" written by Kin and Schaap, the authors provide an exceptional review of the multicellular life cycle and the molecular mechanisms in the dictyostelid aggregative amoebae (cellular slime molds). They provide a clear review of the overall significance of why we should study dictyostelids along with a general context of their evolutionary history. I find this manuscript to be outstandingly well written, clear, and generally helpful for the audience. I am not an expert in the molecular mechanisms that dictyostelids, so I find the way that this manuscript is presented provides me with resource in which my lab can use. I only have 2 minor comments. 

Paragraph 2: I would urge the authors to consider the vast array of aggregative multicellular organisms (mostly amoeboid). A good resource may be (Brown MW, Silberman JD. 2013. The Non-Dictyostelid Sorocarpic Amoebae*. In Romeralo, Escalante, Baldauf (Eds.) Dictyostelids - Evolution, Genomics and Cell Biology. Springer, Heidelberg Germany. pp 219-242. ISBN 978-3-642-38487-5. DOI: 10.1007/978-3-642-38487-5_12.)\

L151: myxogastrids. I would suggest that the authors use the term Myxomycetes. In actuality, Myxogastrids are indeed the plasmodial slime molds, but they are sister to the Ceratiomyxa+Protosporangiid Protosteloid amoebae, together Myxogastria + this group make up the Myxomycetes. The Myxomycetes are sister to the Dictyostelia, technically not the Myxogastria. I understand the in the tree of Figure 2 there may not be the ceratiomyxiids (i.e., Protosporangiida), but I would still suggest this edit. See Kang et al. 2017

Reviewer 2 Report

The manuscript “Evolution of Multicellular Complexity in the Dictyostelid Social Amoebas” provides a very nice review of the evolution of multicellular complexity in a group of social amoebae, especial with respect to the origin of cell types in the context of co-option of stress-responses already present in their unicellular ancestors. In addition, it makes references to similar principles during the evolution of other multicellular lineages. The manuscript is very well written and the figures are clear/useful and relevant.

Minor edits/suggestions/comments:

  • Line 30: Add Herron, M.D.; Borin, J.M.; Boswell, J.C.; Walker, J.; Chen, I.K.; Knox, C.A.; Boyd, M.; Rosenzweig, F.; Ratcliff, W.C. De novo origins of multicellularity in response to predation. Sci Rep 2019, 9, 2328. ??
  • Line 47: “amoeba” should be “amoebas”?
  • Line 77: “from temporal into spatial control” – add Mikhailov, K. V et al. 2009. The origin of Metazoa: a transition from temporal to spatial cell differentiation. Bioessays 31, 758–768; Nedelcu, A.M.; Michod, R.E. The evolutionary origin of an altruistic gene. Mol Biol Evol 2006, 23, 1460-1464
  • Line 94: Add “a” before “multicellular aggregate”?
  • Line 108: add “multicellular” – “over multicellular development”
  • Line 110: “initial fertilization event” – are there distinct mating types?
  • Line 129-131: “The presence of the encystation cycle in non-Group 4 species as well as in other Amoebozoan species suggests that it is an ancestral trait for Dictyostelia, but lost in the Group 4 Dictyostelids.“ Any suggestions/speculations as to why cysts were lost?
  • Lines 146-8: sentence is missing something? – “Amoebozoa in turn members of the phylogenetic supergroup Amorphea, together with Holozoa (Metazoans + unicellular relatives) and Fungi [35] (Fig. 2)”…???
  • Line 163: “loacted” should be “located”
  • Lines 191-2: “why did they bother to make another resistant cell type, spores, through multicellular development?”… not sure this is the best way (although I understand the point) to convey an evolutionary process….
  • Line 230: “Combined, these observations suggest that Dictyostelium sporulation in multicellular fruiting bodies was an adaptation to cold climate”. Whereas the fact that spores in group 4 are more resistant to frost than cysts and spores in the other 3 groups argues that the adaptive role of spores can be related to cold, it is not clear why the evolution of multicellularity was required for the origin of such spores; that is, why not evolve a cold-induced “spore” programme as a single cell individual amoeba? Why aggregate, and form a stalk? Does the stalk provide a benefit in the context of cold temperature? Similarly, if multicellular fruiting bodies evolved as adaptations to cold, why are they also present in group 1-3 that are not forming frost-resistant spores and are not found in cold regions. A more explicit treatment of this hypothesis would help the reader understand the direct connection between the evolution of multicellularity (not just of spores) and cold…
  • Line 232: “multicellular spores” – are the spores multicellular?
  • Line 236-8: “It is well possible that the metabolites produced by this sacrifice are taken up by the prespore cells and used to synthesize their thick spore walls.” Is there any evidence, in this or other system, that autophagy specifically (not other forms of programmed cell death) releases metabolites? If so, such reference(s) should be included, as generally, autophagy is a process that provides nutrients/metabolites to the cell itself, not to other cells…
  • Line 246-9: “cAR mediated induction of spore differentiation is also specifically inhibited in mutants with defective autophagy [57]. In contrast to the stalk cells, the pre-spore cells do not actually show much autophagy [58]. It therefore appears that dictyostelid amoebas will form spores, only if the stalk population undergoes autophagy. “ I am not sure this is the only possible explanation. It seems also possible that the defective autophagy gene has also an effect on spore differentiation. In other words, the same gene that is involved in inducing autophagy in stalk cells is also involved in spore differentiation – ie, it has a pleiotropic effect on two traits/pathways; genes with such effects have been reported in D. discoideum …
  • Line 249-51: “In this scenario, the stalk cells not only function to lift the spores, but also to feed them, typically a major role for somatic cells in multicellular organisms.” Again, it would be useful to provide some suggestion as to how this “feeding” would happen. Also, I am not sure the second part of the sentence is really valid; generally, the major role of somatic cells in multicellular organisms is not really in feeding (but survival – in general)….
  • Line 264: “chemoattract” should be “chemoattractant”?
  • Line 275: “Amobo” should be “Amoebo-’”
  • Line 292: “The transition from slug migration to fruiting body formation is inhibited by ammonia, which is produced by autophagy in the starving cells.”; maybe add “and released in the slug”? Add reference?
  • Line 360: “set aside a population of non-prespore cells for later support cell differentiation” – something is missing?
  • Line 423: Add: Konig SG and AM Nedelcu. 2020. The genetic basis for the evolution of soma: Mechanistic evidence for the co-option of a stress-induced gene into a developmental master regulator. Proceedings of the Royal Society B 287: 20201414.??
  • Line 440: Add: Schlichting, C.D. 2003. Origins of differentiation via phenotypic plasticity. Evol. Dev. 5, 98–105; Nedelcu AM and RE Michod. 2020. Stress responses co-opted for specialized cell types during the early evolution of multicellularity. BioEssays 42(5): 2000029.??
  • Line 448: Add: Sebé-Pedrós, A. et al. 2018. Early metazoan cell type diversity and the evolution of multicellular gene regulation. Nat. Ecol. Evol. 2, 1176–1188??
  • Line 464-5: “explaining its slow evolution into morphological complexity” – should it be “in terms of morphological complexity”?
